# The Effect of Intraocular Pressure-Lowering Medication on Metastatic Uveal Melanomas

**DOI:** 10.3390/cancers13225657

**Published:** 2021-11-12

**Authors:** Jan Pals, Hanneke W. Mensink, Erwin Brosens, Robert M. Verdijk, Nicole C. Naus, Dion A. Paridaens, Emine Kilic, Wishal D. Ramdas

**Affiliations:** 1Department of Ophthalmology, Erasmus University Medical Center, 3015 CA Rotterdam, The Netherlands; j.pals@erasmusmc.nl (J.P.); n.naus@erasmusmc.nl (N.C.N.); a.paridaens@erasmusmc.nl (D.A.P.); e.kilic@erasmusmc.nl (E.K.); 2The Rotterdam Eye Hospital, 3011 BH Rotterdam, The Netherlands; H.Mensink@oogziekenhuis.nl; 3Department of Clinical Genetics, Erasmus University Medical Center, 3015 CA Rotterdam, The Netherlands; e.brosens@erasmusmc.nl; 4Department of Pathology, Section Ophthalmic Pathology, Erasmus University Medical Center, 3015 CA Rotterdam, The Netherlands; r.verdijk@erasmusmc.nl; 5Department of Pathology, Leiden University Medical Center, 2333 ZA Leiden, The Netherlands

**Keywords:** glaucoma, uveal melanoma, intraocular pressure, intraocular pressure-lowering medication, metastatic uveal melanoma

## Abstract

**Simple Summary:**

The most lethal tumor in the eye is metastatic uveal melanomas, while the most common cause of irreversible blindness is glaucoma. Glaucoma is treated by prescribing intraocular pressure-lowering drugs. Theoretically, these drugs may affect the risk of metastasis of intraocular tumors (uveal melanomas). Using data of a long-running and ongoing study on uveal melanomas, we found that eye drops that lower the intraocular pressure by stimulating outflow of fluid (aqueous humor) may increase the risk of metastasis, and subsequent mortality. Therefore, in patients at risk or suspect for uveal melanoma, we recommend choosing ophthalmic drugs with a working mechanism that is not based on the increase of outflow of aqueous humor from the eye.

**Abstract:**

Background: There has been speculation that IOP-lowering medication, which increases aqueous humor outflow, increases the risk of metastatic uveal melanoma (UM). This hypothesis has not been studied previously but is relevant for UM patients who use IOP-lowering medication. The aim of the current study is to assess the association between the use of intraocular pressure (IOP)-lowering medication and the risk of metastatic UM, and mortality. Methods: A retrospective cohort study, in which patients from the Rotterdam Ocular Melanoma Study were included from 1986 onwards. Medical records were evaluated for use of IOP-lowering medication at baseline (i.e., before diagnosis). For each IOP-lowering medication, we divided patients into two groups for comparison (e.g., patients with alpha2-agonist use and patients without alpha2-agonist use). All patients underwent regular ophthalmic examinations and routine screening for metastasis. Survival analyses were initiated to compare groups in each IOP-lowering medication group. In addition, secondary analyses were performed to examine the association between IOP and the development of metastatic UM, and mortality. Results: A total of 707 patients were included of whom 13 patients used prostaglandin or pilocarpine at baseline. For alpha2-agonist, beta-blocker, carbonic anhydrase inhibitor, and oral IOP-lowering medication these were 4, 14, 11, and 12 patients, respectively. The risk of metastatic UM (choroid and ciliary body melanoma) among the prostaglandin/pilocarpine users was significantly higher than controls (HR [95% CI]: 4.840 [1.452–16.133]). Mortality did not differ significantly among the IOP-lowering medications groups, except for the prostaglandin or pilocarpine group (HR [95% CI]: 7.528 [1.836–30.867]). If we combined all IOP-lowering medication that increase aqueous humor outflow, the risk (HR [95% CI]) of metastatic UM and mortality was 6.344 (1.615–24.918) and 9.743 (2.475–38.353), respectively. There was an association between IOP and mortality, but not for the onset of metastatic UM. Conclusion: The use of topical prostaglandin or pilocarpine may increase the risk of metastatic UM and mortality compared to patients without prostaglandin or pilocarpine use. Therefore, use of IOP-lowering medication which increases aqueous humor outflow, should be avoided in patients with (presumed) UM.

## 1. Introduction

Uveal melanoma (UM) is one of the few eye diseases, which has a lethal prognosis in up to 50% of the patients. UM can be classified into choroidal melanoma (CM; including 90% of all UM), ciliary body melanoma (CBM), and iris melanoma (IM) [1]. Annually, in the United States alone, 5.1 per million cases of UM are diagnosed. This incidence causes UM to be the most common primary intraocular malignancy in adults [2]. Up to 50% of UM patients will develop metastatic uveal melanoma (MUM), with a median survival of 12.5 months [3,4]. Fundamentally, UM have a purely hematogenous spread, as there are no lymphatic vessels draining from the eye [4,5,6]. The most common site of metastasis for UM is the liver (93%) [7,8,9]. The pathogenesis of UM to MUM is multifaceted, and is associated with various risk factors (i.e., monosomy 3 and gene mutations) [10].

Occasionally, UM can result in secondary glaucoma [11,12]. Currently, intraocular pressure (IOP) is the major risk factor for glaucoma. One of the most common treatments to lower IOP is the use of eye drops, of which prostaglandin analogues are the most potent ocular hypotensive medications used by ophthalmologists [13].

In general, there are two groups of IOP-lowering medications; medications that reduce the IOP by increasing aqueous humor outflow (e.g., prostaglandins, pilocarpines, alpha2-agonists) and medications that reduce the IOP by reducing the production of aqueous humor from the ciliary body (e.g., alpha2-agonists, beta-blockers, (oral) carbonic anhydrase inhibitors) [14,15,16,17,18,19]. Regarding the first group, it has been hypothesized that these medications have a higher risk of metastatic disease as a result of the increased outflow of aqueous humor. Nonetheless, reports on IOP-lowering medication use in UM patients and the risk for MUM are scarce. Moreover, the scientific evidence supporting this hypothesis is poor and consists of only a few case reports. One case report described an iris melanoma with secondary glaucoma treated with latanoprost. However, no mention was made of the risk of metastasis using latanoprost [20]. Furthermore, the hypothesis that IOP-lowering medication, which increases aqueous humor outflow, increases the risk of MUM, has not been studied previously.

Therefore, we performed a retrospective study to compare the risk of clinically detectable metastasis in UM patients between various groups of IOP-lowering medication. In addition, we examined whether IOP-lowering medication that increases aqueous humor outflow is associated with a higher mortality. Finally, we assessed whether elevated IOP is associated with MUM and mortality.

## 2. Material and Methods

### 2.1. Study Population

We performed a retrospective study with data from the Rotterdam Ocular Melanoma Study (ROMS). ROMS is an ongoing study and is a collaboration between the department of ophthalmology, pathology and clinical genetics of the Erasmus University Medical Center and The Rotterdam Eye Hospital, both in Rotterdam, the Netherlands. Data of all patients with UM were collected during each visit from 1986 onwards. The following data were extracted from the database: age, gender, ethnicity, tumor diagnosis (CM, CBM or IM), medical history, visual acuity, IOP, medication use (both ocular and systemic), treatment (e.g., enucleation, brachytherapy, stereotactic radiotherapy), TNM classification (8th edition), tumor genotyping, tumor histology, date of development of metastasis (i.e., clinically detectable MUM), and date of death (if applicable).

Only patients with a diagnosis of CM or CBM were included in this study. Patients with IM were excluded, as one of the important parameters in suspecting an IM is increased IOP [21]. All IOP-lowering medications had to be prescribed before diagnosis of UM. Thus, patients who were treated for secondary glaucoma due to UM were excluded. Patients with uncategorized tumor diagnosis (i.e., CM or CBM) were excluded to be sure that only true CM or CBM were in those subgroups. For the (pooled) UM group analyses the type of diagnosis (CM or CBM) is not necessary to know; therefore, we did include these patients in the (pooled) UM group if their iris was not suspect. The Medical Ethics Committee of the Erasmus University Rotterdam had approved the study protocol (registration number MEC 2009–375) and all participants had given written informed consent in accordance with the declaration of Helsinki.

### 2.2. Assessment of Exposure

IOP-lowering medication were categorized into different groups (based on their clinical effect): pilocarpine and prostaglandin-analogues (PP) were merged into one group, based on their similar working mechanism (increasing aqueous humor outflow) and to ensure sufficient case numbers in the study group, as our dataset contained only one case of pilocarpine in total. In this analysis PP users were compared with non-PP users. Non-PP users were defined as all remaining patients.

Alpha2-agonists were categorized separately, because the working mechanism is twofold: reducing IOP by decreasing production of aqueous humor and by increasing the outflow of aqueous humor [17]. Other IOP-lowering medications, that reduce the production of aqueous humor from the ciliary body, were categorized separately as well (e.g., beta-blockers, carbonic anhydrase inhibitors (CAI) and oral IOP-lowering medication). Non-alpha2-agonist users, non-beta-blocker users, non-carbonic anhydrase inhibitor (non-CAI) users and non-oral IOP-lowering medication users were defined similarly as non-PP users. Subsequently, alpha2-agonist users were compared with non-alpha2-agonist users, beta-blocker users with non-beta-blocker users, CAI users with non-CAI users, and oral IOP-lowering medication users with non-oral IOP-lowering medication users.

### 2.3. Assessment of Main Outcomes

Patients were routinely (each 6 months) screened for the development of metastasis by blood tests (i.e., liver function enzymes) and/or ultrasound of the liver (from 2010 onwards). A computer tomography (CT) scan and/or magnetic resonance imaging (MRI) scan of the liver was performed in patients with MUM. Furthermore, for all patients that underwent enucleation or secondary endoresection as the treatment for UM, immunohistochemistry staining (to detect BAP1 expression) and genotyping (including *BAP1*- and *SF3B1-*mutations) was performed on the tumor tissue. If immunohistochemistry staining was negative (i.e., loss of BAP1 expression), we assumed a *BAP1*-mutation. Details on the immunohistochemistry and genotyping methods are described elsewhere [10,22,23]. Mortality was assessed by date of death.

### 2.4. Statistical Analyses

General baseline characteristics were compared among groups using independent t-tests for continuous variables (or Mann–Whitney U test if not normally distributed) and chi-square tests (or Fisher’s exact test if applicable) for categorical variables, for each medication group.

First, we assessed the differences in IOP-lowering medication, metastasis, and mortality between CM and CBM. To assess metastasis and mortality rate for each IOP-lowering medication group we applied Kaplan–Meier survival analyses. This was done for CM, CBM, and both combined (UM). The differences between the curves were analyzed using the log-rank-test. Cox proportional hazard regression models were performed to calculate the hazard ratio (HR) with corresponding 95% confidence intervals (CI) for metastatic choroidal melanoma (MCM), metastatic ciliary body melanoma (MCBM), and MUM (i.e., pooled) for each medication group. The model was adjusted for age, gender, IOP, and significant baseline characteristics (i.e., variables with *p* < 0.05 in the univariate comparisons). All models were adjusted for the same covariates within each type of UM.

A second Cox proportional hazard regression model was performed to examine the mortality for each medication group. This model was adjusted identical to the Cox proportional hazard regression model for metastasis. Finally, as a secondary outcome, we analyzed the association between IOP and both the onset of MUM and mortality, using univariate and multivariate Cox proportional hazard regression models. The multivariate Cox regression model was adjusted identical to the previous Cox regression models.

Baseline was defined as the date of diagnosis with UM (i.e., CM or CBM). For the IOP-lowering medication analyses, all patients with a baseline date prior to the date the IOP-lowering medication was launched in the Netherlands (e.g., alpha2-agonists were launched in 2000) were excluded. The dates pilocarpine, prostaglandins, beta-blockers, carbonic anhydrase inhibitors, and oral IOP-lowering medication were launched, were 1875, 1996, 1978, 1994, and 1954, respectively [24]. The Cox regression models for the association between IOP and the onset of metastasis and mortality included all patients from 1986 onwards. Time-to-event (i.e., follow-up) was counted from baseline till the date of either metastasis or mortality. If the size of an IOP-lowering medication group was ≤5, we considered the number of medication users too small to perform a meaningful analysis. Additional analyses were performed in which IOP-lowering medications that affect the outflow of aqueous humor (prostaglandins, pilocarpines, and alpha2-agonists) were merged into one group and compared to non-users, as a secondary outcome on our hypothesis. Similarly, we merged IOP-lowering medication with an effect on the production of aqueous humor (i.e., alpha2-agonists, beta-blockers, CAI, and oral IOP-lowering medication) in another group. Statistical analyses were performed using SPSS v22.0 for Windows (SPSS Inc., Chicago, IL, USA). *p*-values of <0.05 were considered statistically significant.

## 3. Results

A total of 707 patients with UM were included (570 [80.6%] with CM). Of these, 239 UM patients developed metastasis of whom 208 (87.0%) deceased. For three (1.3%) patients, metastatic disease was not associated with mortality, one (0.4%) patient was lost-to-follow-up, and the remaining 27 (11.3%) patients were alive at the end of the follow-up. The use of PP and oral IOP-lowering medication in CM patients was significantly lower than in CBM patients (Table 1). As expected, compared to CM patients those with a CBM had a significantly higher risk to develop metastasis, but mortality rates were similar (Appendix A).

Univariate variables that were used for adjustment for all IOP-lowering medications in the Cox proportional hazard regression model were as follows: age, gender, IOP, IOP-lowering medication use, and tumor treatment (Appendix A).

### 3.1. Choroidal Melanoma

#### 3.1.1. PP-Users

For the PP analyses, 521 patients were included from 1996 onwards, of which 6 (1.2%) patients used PP at baseline (Appendix A). Of these, one patient used pilocarpine. There was a significant difference in time till development of MCM between the PP users and the non-PP users (median [interquartile range (IQR)]: 1.13 years [0.99–2.22] and 4.12 years [1.97–8.85]; *p =* 0.013, respectively) and in the total follow-up time (i.e., till death) (median [IQR] 1.98 years [1.24–2.76] and 4.58 years [2.25–8.87]; *p* = 0.028, respectively). The Kaplan–Meier cumulative incidence curve showed a significant difference for MCM and mortality between the two groups (Appendix A; log-rank *p* < 0.001 and log-rank *p* = 0.028, respectively). Patients who used PP had a significantly higher risk of developing MCM (HR [95% CI]: 11.413 [2.773–46.968]) and of mortality (HR [95% CI]: 10.367 [2.424–44.342]; Table 2A) compared to non-PP users.

#### 3.1.2. Alpha2-agonists

A total of 474 patients were included from 2000 onwards, of which 3 (0.6%) patients used alpha2-agonists. As the number of alpha2-agonist users was ≤5, we did not analyze this group for CM.

#### 3.1.3. Beta-Blockers, CAI, and Oral IOP-Lowering Medication

Baseline characteristics for beta-blockers-, CAI-, and oral IOP-lowering medication users are shown in Supplementary Appendix A, respectively. For beta-blockers significant differences were found in Kaplan–Meier cumulative incidence curve and HR for MCM, but not for mortality (Appendix A and Table 2A). For CAI- or oral IOP-lowering medication users, no significant differences were found in Kaplan–Meier cumulative incidence curves and HRs for MCM and mortality (Appendix A–H, and Table 2A, respectively).

### 3.2. Ciliary Body Melanoma

Unfortunately, all IOP-lowering medication groups had a size ≤ 5 (Table 1). Therefore, we did not analyze the data for CBM.

### 3.3. Uveal Melanoma (CM and CBM Pooled)

#### 3.3.1. PP-Users

In the total UM group 642 patients were included from 1996 onwards, of which 13 (2.0%) patients used PP at baseline (Appendix A). As expected, patients using PP had a higher IOP than those without the use of PP at the date of UM diagnosis. There was a significant difference in time till development of MUM between the PP users and the non-PP users (median [IQR] 1.24 years [0.86–2.22] and 4.08 years [1.80–8.46]; *p* = 0.004, respectively) and in the total follow-up time (i.e., till death) (median [IQR] 1.51 years [0.95–2.76] and 4.44 years [2.03–8.53]; *p =* 0.004, respectively).

Figure 1A shows differences in cumulative incidence for MUM between the PP users and non-PP users. There was a significant difference for MUM between the two groups (log-rank *p* = 0.023). However, no significant difference was found in mortality between both groups (log-rank *p* = 0.463; Figure 1B). Patients who used PP had a significantly higher risk of developing MUM and mortality compared to non-PP users (HR [95% CI]: 4.840 [1.452–16.133] and HR [95% CI]: 7.528 [1.836–30.867], respectively; Table 2B).

#### 3.3.2. Alpha2-agonists

A total of 589 patients were included from 2000 onwards, of which 4 (0.7%) patients used alpha2-agonists (Appendix A). As the number of alpha2-agonist users was ≤5, we did not analyze this group for UM.

#### 3.3.3. Beta-Blockers, CAI, and Oral IOP-Lowering Medication

Baseline characteristics for beta-blockers-, CAI-, and oral IOP-lowering medication users are shown in Supplementary Appendix A, respectively. None of these medication groups had a significant effect on cumulative incidence or HRs for MUM and mortality (Figure 1D–H, and Table 2B, respectively), with the exception of the Kaplan–Meier cumulative incidence curve in beta-blocker users for MUM (Figure 1C).

### 3.4. IOP-Lowering Medications That Increase Aqueous Humor Outflow and IOP-Lowering Medications That Reduce the Production of Aqueous Humor

For IOP-lowering medication that increases the outflow of aqueous humor, the cumulative incidence curves for MUM were significantly higher compared to non-users, but not for mortality (Figure 2A,B, respectively). Additionally, the risk of MUM and mortality were significantly higher (HR [95% CI]: 6.344 [1.615–24.918] and HR [95% CI]: 9.743 [2.475–38.353], respectively). For IOP-lowering medication that reduces the production of aqueous humor, the cumulative incidence curves for MUM and mortality were similar to non-users (Figure 2C,D, respectively). Furthermore, they did not affect the risk of MUM (HR [95% CI]: 0.481 [0.127–1.814]), though an association with mortality (HR [95% CI]: 0.143 [0.030–0.686]) was found.

### 3.5. IOP and Risk of Metastasis and Mortality

IOP was not associated with either MUM or mortality in the univariate Cox regression models (HR [95% CI]: 1.008 [0.988–1.028] and 1.012 [0.996–1.028], respectively). Additionally, in the multivariate Cox regression model IOP was not significantly associated with MUM (HR [95% CI]: 1.007 [0.983–1.032]); however, IOP was associated with a slightly higher mortality (HR [95% CI]: 1.023 [1.005–1.042]).

## 4. Discussion

To the best of our knowledge, this is the first study investigating the impact of IOP-lowering medication use on the risk for MUM and mortality. We found that, independent of IOP, the use of PP in patients with UM is associated with a significantly earlier onset of MCM and MUM, compared to those not using PP.

It is interesting that the PP group, despite its small sample size in CM (n = 6) and UM (n = 13; CM+CBM), still resulted in a significant HR for development of MCM, MUM, and mortality. It is to be expected that a longer follow-up time results in a higher probability to detect an event, either MUM or mortality. Although the PP users had a shorter follow-up time compared to the non-PP users, the occurrence of MCM and MUM was still significantly higher in the PP users. Unfortunately, the number of alpha2-agonist users was too low to perform meaningful analyses. If we disregarded that the number of alpha2-agonist users was ≤5, we would have found a non-significant high HR for MUM and mortality (data not shown). Furthermore, the HRs increased when we combined all IOP-lowering medications that increase aqueous humor outflow. This finding supports our hypothesis that the use of IOP-lowering medications that increase aqueous humor outflow may increase the risk for MUM and mortality, as alpha2-agonists have an effect on both aqueous humor outflow and production [17].

In general, ophthalmologists use prostaglandin F2α to reduce IOP, however, prostaglandin E2 has a similar uveoscleral outflow effect [25,26]. As we mentioned in Section 1, UM have a purely hematogenous spread. There are several possible mechanisms that support our findings of the increased risk for metastasis (and therefore mortality) due to prostaglandin use. First, prostaglandin (F2α and E2) exposure stimulates vascular endothelial growth factor (VEGF) and therefore increases angiogenesis and vascularization [27,28,29,30]. Second, prostaglandin F2α and E2 increase the scleral- and vascular permeability, respectively [31,32]. Third, in rats it has been reported that prostaglandin E2 exposure increases fundus blood flow and vasodilatation [33]. Furthermore, prostaglandin E2 is associated with carcinogenesis and tumor progression, including melanoma [34,35].

In the current study, IOP-lowering medication that reduces the production of aqueous humor was neither associated with metastasis nor mortality, with the exception of the increased risk of MCM in beta-blocker users. The latter is in contrast to Bustamante et al. who suggested the opposite effect, as propranolol may have a potent anti-tumor effect in UM cells in vitro [36]. However, our result is questionable as the association was only present in CM and not in UM. Moreover, the HRs decreased when we combined all IOP-lowering medications that decrease aqueous humor production (including beta-blockers and CAI), particularly the HR for mortality (Table 3). A large part of all IOP-lowering medications that decrease aqueous humor production consist of CAI and oral IOP-lowering medications (56.1%), both are fundamentally similar. Carbonic anhydrase expression is associated with a high malignancy grade in tumor endothelium, as a result CAI might reduce tumor growth and neoangiogenesis [37,38]. This could clarify our significantly low HR for mortality in the merged IOP-lowering medication group that decrease aqueous humor production.

The IOP itself was not associated with the risk to develop MUM, but the multivariate Cox regression model suggested a slightly higher risk of mortality. This effect seemed to be independent of the use of IOP-lowering medication. However, if we excluded all patients on IOP-lowering medication, IOP was neither associated with MUM nor with mortality (HR [95%CI]: 0.992 [0.963–1.023] and 1.007 [0.988–1.026], respectively). Therefore, it is not likely that confounding-by-indication affected our findings.

It should be noted that the risk of metastasis highly depends on genetic mutations (*BAP1*, *SF3B1*, and *EIF1AX*) in the tumor [10,22]. Of these, *EIF1AX* is not associated with a reduced survival in UM patients [10]. To assess the impact of the underlying genetic mutations on the risk of metastasis, we assessed *BAP1*- and *SF3B1*-mutations in patients with available tumor tissue. However, we found no differences in *BAP1-* and *SF3B1-*mutation status or TNM-classification (tumor size) between the IOP-lowering medication groups and the control groups. Moreover, if we adjusted the Cox regression models additionally for tumor size and BAP1, the results did not alter (data not shown). In the PP group 100% of patients had a *BAP1*-mutation; while in the control group this was 66%. Albeit, statistically not significant (*p* = 0.305; Appendix A), yet still the few PP users might simply by chance belong to an intrinsically high-risk group. As a matter of fact, almost all *BAP1* mutated UM develop metastases. Nevertheless, if we excluded all patients without a *BAP1*-mutation (resulting in a smaller control group but also resulting in 100% of both PP users and non-users having a *BAP1*-mutation), the results become even stronger (Appendix A). This suggests that non-genetic factors might be involved as well in the time-of-onset and risk of metastasis.

There are several other possible prognostic factors for UM such as presence/absence of vascular invasion in UM samples, presence/absence of extraocular extension, and tumor infiltrating lymphocytes. Previously, we assessed these variables in a subset; nonetheless, these factors did not affect the survival in our UM-patients [6]. Unfortunately, we do not have data on the melanin pigmentation. However, the melanin content in UM affects the outcome in patients treated with radiotherapy. In the current study, there was no significant difference in number of patients treated with radiotherapy between patients using IOP-lowering medication and non-users.

Due to the retrospective design our study has several limitations. We did not consider the duration and frequency of IOP-lowering medication use. As UM could be associated with an increased IOP, we included only patients who started using IOP-lowering medication prior to UM diagnosis and all iris melanomas were excluded. All available treatments for UM have the same worse prognosis. Even patients who undergo enucleation may still develop metastasis. This might be caused by micrometastasis (clinically not detectable) that left the eye before the patient underwent any UM treatment/enucleation. Second, a disadvantage of using Cox regression in the current study is the included period at-risk. As mentioned in Section 2, the earliest baseline date was 1986, while at that moment there was only pilocarpine, beta-blockers, and oral IOP-lowering medication available for the treatment of elevated IOP. In the Netherlands, the first prostaglandin-analogues were launched in 1996 [24,39]. Therefore, we performed analyses in which we excluded all patients with a baseline date prior to 1996; as a matter of fact this resulted in a shorter follow-up duration. Third, we compared the IOP-lowering medication vs. non-users of the respective medication. Thus a few patients from the control group were on other IOP-lowering medication; however, excluding these patients from the control groups did not alter the results (data not shown). Fourth, due to the few published case reports it is likely that an ophthalmologist will not prescribe prostaglandin-analogues as a first choice treatment option. These case reports were published after 2009 [20,40,41]. Nevertheless, in the current study the baseline dates of the prostaglandin-analogues user were 2004 till 2019. Finally, UM is a scarce disease, let alone the combination with IOP-lowering medication, making it less suitable for a prospective study design. Moreover, treating UM patients with medication that have been hypothesized to increase the risk of metastasis would probably not be acceptable from an ethical point of view.

## 5. Conclusions

The current results suggest that the use of PP (or IOP-lowering medications that increase aqueous humor outflow) to lower IOP in UM patients may increase the risk of MUM and subsequently mortality. Therefore, we recommend being cautious when choosing IOP-lowering medications that increase aqueous humor outflow in patients with or suspect for UM. Due to the low prevalence of UM, the number of patients with both UM and IOP-lowering medication was very low in the present study. Hence, future studies are required to strengthen our current conclusions and to elucidate the possible underlying mechanisms.

## Figures and Tables

**Figure 1 cancers-13-05657-f001:**
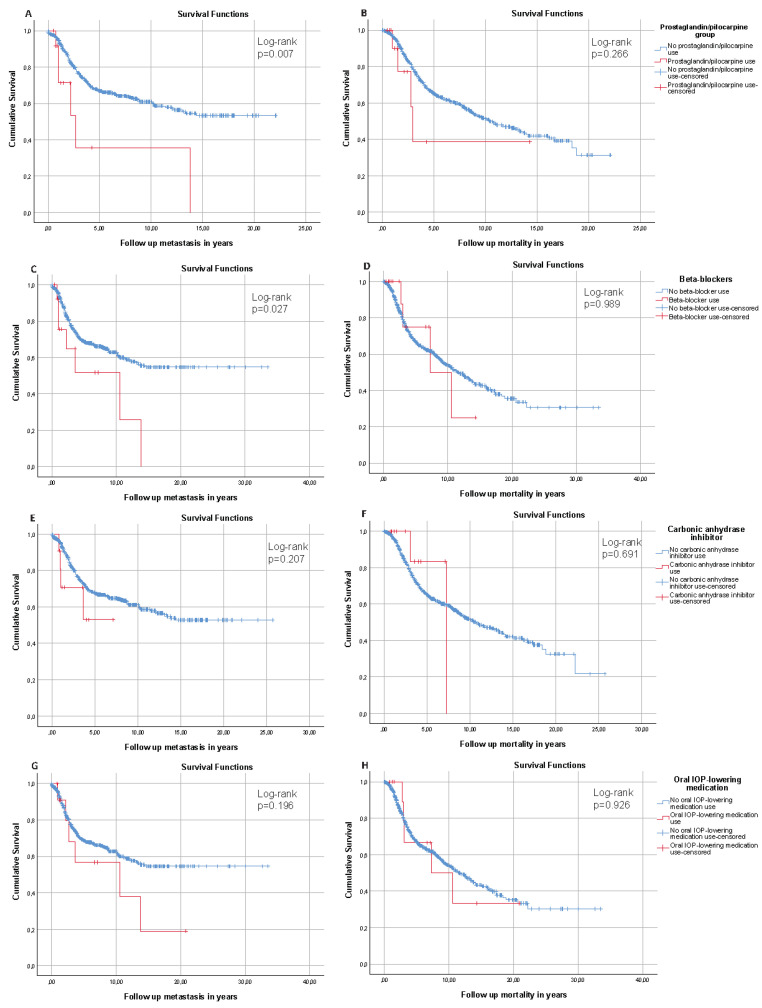
Kaplan–Meier cumulative incidence curves of metastasis (**left**) and mortality (**right**) in uveal melanoma patients with and without prostaglandin/pilocarpine use (**A**,**B**), with and without beta-blocker use (**C**,**D**), with and without carbonic anhydrase inhibitor use (**E**,**F**), and with and without oral intraocular pressure-lowering medication use (**G**,**H**).

**Figure 2 cancers-13-05657-f002:**
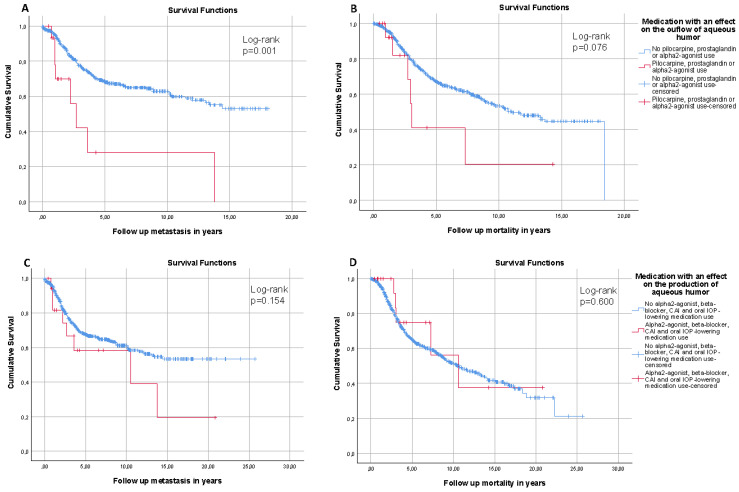
Kaplan–Meier cumulative incidence curves of metastasis (**left**) and mortality (**right**) in uveal melanoma patients for IOP-lowering medications that increase aqueous humor outflow (**A**,**B**), and for IOP-lowering medications that reduce the production of aqueous humor (**C**,**D**).

**Table 1 cancers-13-05657-t001:** IOP-lowering medication use between choroidal- and ciliary body melanoma.

	CM	CBM	CM (Total)	CBM (Total)	*p*-Value
PP	5 Prostaglandin 1 Pilocarpine	3 Prostaglandin 1 Prostaglandin/beta-blocker	6 (1.1)	4 (5.6)	0.018 *
Alpha2-agonist	3 Alpha2-agonist	1 Alpha2-agonist	3 (0.5)	1 (1.4)	0.379
Beta-blocker	3 Beta-blocker 7 Beta-blocker/CAI	1 Beta-blocker 1 Beta-blocker/prostaglandin 1 Beta-blocker/CAI	10 (1.8)	3 (4.2)	0.170
CAI	2 CAI 7 CAI/beta-blocker	1 CAI/beta-blocker	9 (1.6)	1 (1.4)	1.000
Oral	NA	NA	7 (1.2)	4 (5.6)	0.026 *

* = indicates statistical significance (*p* < 0.05); NA = not applicable/available; CM = choroidal melanoma; CBM = ciliary body melanoma; PP = prostaglandin/pilocarpine; CAI = carbonic anhydrase inhibitor; Oral = oral IOP-lowering medication.

**Table 2 cancers-13-05657-t002:** Risk of metastatic choroidal- and uveal melanoma and mortality. Presented as hazard ratio with corresponding 95% confidence interval and *p*-value.

	Risk of Metastasis			Risk of Mortality		
	HR	95% CI	*p*-Value	HR	95% CI	*p*-Value
A. CM ^α^						
-PP	11.413	2.773–46.968	0.001 *	10.367	2.424–44.342	0.002 *
-Beta-blocker	37.725	2.353–604.867	0.010 *	12.834	0.028–5891.620	0.414
-CAI	0.085	0.003–2.060	0.130	0.000	0.000–3.218 × 10^−44^	0.870
-Oral	0.045	0.001–1.477	0.082	0.057	0.000–26.693	0.361
B. UM (CM+CBM) ^β^						
-PP	4.840	1.452–16.133	0.010 *	7.528	1.836–30.867	0.005 *
-Beta-blocker	6.024	0.701–51.786	0.102	1.039	0.094–11.546	0.975
-CAI	0.575	0.064–5.150	0.621	0.000	0.000–7.741 × 10^−100^	0.933
-Oral	0.112	0.010–1.204	0.071	0.234	0.025–2.201	0.204

* = indicates statistical significance (*p* < 0.05); α = adjusted for age, gender, IOP, IOP-lowering medication, and tumor treatment (Appendix A); β = adjusted for age, gender, IOP, IOP-lowering medication, tumor diagnosis, and tumor treatment (Appendix A); PP = prostaglandin/pilocarpine; CAI = carbonic anhydrase inhibitors; Oral = acetazolamide; HR = hazard ratio; CI = confidence interval; CM = choroidal melanoma; CBM = ciliary body melanoma; UM = uveal melanoma.

**Table 3 cancers-13-05657-t003:** Risk of metastatic uveal melanoma and mortality. The baseline for the Cox proportional hazard model included those using medication with an effect on the outflow of aqueous humor and controls from 2000 onwards. For the medication group with an effect on the production of aqueous humor, the baseline was set to 1994. Presented as hazard ratio with corresponding 95% confidence interval and *p*-value.

	Risk of Metastasis			Risk of Mortality		
	HR	95%CI	*p*-Value	HR	95%CI	*p*-Value
Outflow β	6.344	1.615–24.918	0.008 *	9.743	2.475–38.353	0.001 *
Production β	0.481	0.127–1.814	0.280	0.143	0.030–0.686	0.015 *

* = indicates statistical significance (*p* < 0.05); β = adjusted for age, gender, IOP, IOP-lowering medication, tumor diagnosis, and tumor treatment.

## Data Availability

The data presented in this study are available on request from the corresponding author.

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
