# Peer review of "The Effect of Intraocular Pressure-Lowering Medication on Metastatic Uveal Melanomas"

_cancers, 2021, doi:10.3390/cancers13225657_

Round 1

Reviewer 1 Report

The manuscript entitled “The effect of intraocular pressure-lowering medication on metastatic uveal melanomas” addresses the interesting issue of IOP and IOP-treatment as potential tumor progression risk factors. The study is well performed, the limitations are indicated and adequately discussed and the paper is well written.

The authors state in the introduction that “Occasionally, UM can result in secondary glaucoma”. From the text I must assume that IOP treatment was applied to control UM induced secondary glaucoma. It is, however, not clear how this applies to cases with enucleation.

The timing of IOP treatment is not described. I assume that if it is directed at the UM induced secondary glaucoma, IOP treatment started after the diagnosis of UM. If not, this should be clearly stated. Yet if IOP treatment started after UM diagnosis then the presumed role of IOP treatment in metastasis and survival is not clear. Soon after diagnosis, UM is treated in a way to eradicate the tumor and local recurrence is very low. After removal, there is no source of potentially metastasizing cells whose propension to leave the eye could be enhanced by IOP or IOP treatment. In any case, the hypothetical etio-pathological mechanism by which IOP or IOP treatment might affect UM metastasis and/or survival must be discussed.

There is some concern with regard to the statistics. Considering various treatments, a post-hoc test would be required and patients treated in a specific manner should be compared to untreated patients i.e. patients treated with none of the IOP treatments. The fact that the primary UM of 100% of PP treated patients had a BAP1 mutation but only 66% of the untreated is statistically not significant yet still the few PP treated cancers might simply by chance belong to an intrinsically high-risk group. As a matter of fact, almost all BAP1 mutated Ums develop metastases. I recommend to send the paper for statistical review.

Author Response

Reviewer: 1

Comments to the Author

The manuscript entitled “The effect of intraocular pressure-lowering medication on metastatic uveal melanomas” addresses the interesting issue of IOP and IOP-treatment as potential tumor progression risk factors. The study is well performed, the limitations are indicated and adequately discussed and the paper is well written.

The authors state in the introduction that “Occasionally, UM can result in secondary glaucoma”. From the text I must assume that IOP treatment was applied to control UM induced secondary glaucoma. It is, however, not clear how this applies to cases with enucleation.

REPLY and Change: We agree that this is not clear from the manuscript. UM can result in secondary glaucoma before or after UM treatment:

-Before treatment the glaucoma could be caused by iris melanomas. -> Therefore we excluded all iris melanomas.

-After treatment a patient can develop neovascular glaucoma.

In the current study patients did not have a diagnosis of UM at the time of prescription of IOP-lowering medication. Most likely, they used the medication for prevention (in case of many risk factors), ocular hypertension, or primary glaucoma.

Indeed, IOP-lowering medication was stopped in patients who underwent enucleation as a treatment for UM. However, patients are still at-risk for MUM after enucleation. It should be noted that we do not have data on the duration of usage of the IOP-lowering medication (see Discussion-section last paragraph). We made changes in the Methods-section paragraph 2.1.

The timing of IOP treatment is not described. I assume that if it is directed at the UM induced secondary glaucoma, IOP treatment started after the diagnosis of UM. If not, this should be clearly stated. Yet if IOP treatment started after UM diagnosis then the presumed role of IOP treatment in metastasis and survival is not clear. Soon after diagnosis, UM is treated in a way to eradicate the tumor and local recurrence is very low. After removal, there is no source of potentially metastasizing cells whose propension to leave the eye could be enhanced by IOP or IOP treatment. In any case, the hypothetical etio-pathological mechanism by which IOP or IOP treatment might affect UM metastasis and/or survival must be discussed.

REPLY and Change: We totally agree with the reviewer. Therefore, only patients who started using IOP-lowering medication prior to UM diagnosis were included. And, all iris melanomas were excluded. Please note that there is no evidence that enucleation has a better prognosis than SRT or Proton beam therapy. Patient who underwent enucleation may still develop metastasis. This might be because of micro-metastasis caused by malignant cells, which have left the eye before any UM treatment/enucleation.

We have clarified this in the Discussion-section last paragraph.

There is some concern with regard to the statistics. Considering various treatments, a post-hoc test would be required and patients treated in a specific manner should be compared to untreated patients i.e. patients treated with none of the IOP treatments. The fact that the primary UM of 100% of PP treated patients had a BAP1 mutation but only 66% of the untreated is statistically not significant yet still the few PP treated cancers might simply by chance belong to an intrinsically high-risk group. As a matter of fact, almost all BAP1 mutated Ums develop metastases. I recommend to send the paper for statistical review.

REPLY and Change: We agree with the reviewer that the association could still be confounded by the BAP1-mutation. As recommended by the reviewer, we approached a statistician (see Acknowledgement-section). A post-hoc analysis was performed in which we excluded all patients without a BAP1-mutation. This resulted in the following HR (95%CI) for the PP-group:

PP (all CM patients; Table 2)

-Metastasis -> 11.413 (2.773-46.968) p=0.001

-Mortality -> 10.367 (2.424-44.342) p=0.002

PP (only CM patients with BAP1-mutation)

-Metastasis -> 16.868 (3.698-76.946) p<0.001

-Mortality -> 13.271 (3.070-57.364) p=0.001

PP (all CM+CBM patients; Table 2)

-Metastasis -> 4.840 (1.452-16.133) p=0.010

-Mortality -> 7.528 (1.836-30.867) p=0.005

PP (only CM+CBM patients with BAP1-mutation)

-Metastasis -> 7.731 (2.046-29.210) p=0.003

-Mortality -> 9.794 (2.391-40.123) p=0.002

These analyses show that if we include only controls with a BAP1-mutation (which results in a smaller control group), the results become even stronger (in terms of HR’s and p-values). We have added this information to the Discussion-section and for the analyses we added a new table: Supplementary Table S10.

Reviewer 2 Report

This is an interesting retrospective study

The paper for the most part is well written. It has important clinical implications.

In fact I do not find any major weaknesses.

Perhaps authors could include melanin pigmentation of melanoma tumors as an additional variable in the analysis. However this is not necessary but the authors are encouraged to make this additional effort, since melanin content may affect the outcome of therapy (Oncotarget 20:17844-1785 2016 Feb 3. doi: 10.18632/oncotarget.7528; Exp  Dermatol 24: 258-259, 2015).

In general this is an excellent study that will be of strong interest to the readers

Author Response

Reviewer: 2

Comments to the Author

This is an interesting retrospective study

The paper for the most part is well written. It has important clinical implications.

In fact I do not find any major weaknesses.

Perhaps authors could include melanin pigmentation of melanoma tumors as an additional variable in the analysis. However this is not necessary but the authors are encouraged to make this additional effort, since melanin content may affect the outcome of therapy (Oncotarget 20:17844-1785 2016 Feb 3. doi: 10.18632/oncotarget.7528; Exp  Dermatol 24: 258-259, 2015).

In general this is an excellent study that will be of strong interest to the readers

REPLY: We thank the reviewer for his kind comments. Unfortunately, we do not have data on the melanin pigmentation. However, the melanin content in UM affects the outcome in patients treated with radiotherapy. In the current study there was no significant difference in number of patients treated with radiotherapy between patients using IOP-lowering medication and controls. This information has now been added to the Discussion-section.

Reviewer 3 Report

In the manuscript "The effect of intraocular pressure-lowering medication on metastatic uveal melanomas" by Jan Pals et al. the authors aimed to address the important question on whether IOP-lowering drugs, employed to increases aqueous humor outflow, might lead to an enhancement of metastasis development in uveal melanoma (UM). The study comprises a large cohort of patients over a 35-year period. Through well conducted research, the authors found that the use of topical prostaglandin or pilocarpine is associated with an increased risk of metastases development and mortality in uveal melanoma. The paper is well written and organized and the theme is interesting and relevant, with their findings having important implications in the management of patients who are afflicted by this potentially deadly disease. However, there are important aspects that should merit the attention of the authors:

  1. Even though the authors performed a comprehensive comparison between groups in terms of baseline characteristics of patients, such comparison was apparently not performed for important features which have a prognostic impact in uveal melanoma, such as presence or absence of vascular invasion in uveal melanoma samples, presence or absence of extraocular extension, tumour infiltrating lymphocytes (TILs) and tumour infiltrating macrophages (TIMs). It would be important to perform such comparisons to further strengthen the relevance of their important findings.

  1. Since the study comprises a considerable period of 35 years, it would be important to understand if the uveal melanoma cases were all ascribed a TNM classification according to the most recent 8th Edition of the AJCC cancer staging manual.

  1. Since the study comprises a considerable period of 35 years, the authors should more thoroughly discuss the evolution of usage of IOP-lowering drugs in the treatment of uveal melanoma patients; and also changes in the follow-up of these patients.

  1. The discussion section would also benefit from a better initial paragraph, as well as, a better organization of the discussion flow.

  1. In the supplementary material table the authors should clarify the presence of p-values of 1.000 in comparisons involving groups which have 0 patients displaying a specific feature (for example, positive family history for UM in table S1). Does it make sense to report a p-value in these cases?

Therefore, if considered relevant for publication in Cancers, the authors will have to address the aspects highlighted above before the final format of publication is achieved.

Author Response

Reviewer: 3

Comments to the Author

In the manuscript "The effect of intraocular pressure-lowering medication on metastatic uveal melanomas" by Jan Pals et al. the authors aimed to address the important question on whether IOP-lowering drugs, employed to increases aqueous humor outflow, might lead to an enhancement of metastasis development in uveal melanoma (UM). The study comprises a large cohort of patients over a 35-year period. Through well conducted research, the authors found that the use of topical prostaglandin or pilocarpine is associated with an increased risk of metastases development and mortality in uveal melanoma. The paper is well written and organized and the theme is interesting and relevant, with their findings having important implications in the management of patients who are afflicted by this potentially deadly disease. However, there are important aspects that should merit the attention of the authors:

Even though the authors performed a comprehensive comparison between groups in terms of baseline characteristics of patients, such comparison was apparently not performed for important features which have a prognostic impact in uveal melanoma, such as presence or absence of vascular invasion in uveal melanoma samples, presence or absence of extraocular extension, tumour infiltrating lymphocytes (TILs) and tumour infiltrating macrophages (TIMs). It would be important to perform such comparisons to further strengthen the relevance of their important findings.

REPLY and Change: We agree with the reviewer that this information may add more information about the prognostic impact of UM. Unfortunately, we have no information on the vascular invasion and TIMs; however, we have data on vascular loops and TILs in a subset of patients. Nonetheless, both variables did not affect the survival in UM-patients.[Van Beek, et al. 2019] Therefore, it is not likely that further adjustment of our analyses for these variables would have affected our results. This information has now been added to the Discussion-section.

Van Beek JGM, et al. Absence of Intraocular Lymphatic Vessels in Uveal Melanomas with Extrascleral Growth. Cancers (Basel). 2019 Feb 15;11(2):228.

Since the study comprises a considerable period of 35 years, it would be important to understand if the uveal melanoma cases were all ascribed a TNM classification according to the most recent 8th Edition of the AJCC cancer staging manual.

REPLY and Change: We are sorry for not mentioning this information. All uveal melanoma were classified using TNM classification of the 8th edition. We have added this information to the Methods-section paragraph 2.1.

Since the study comprises a considerable period of 35 years, the authors should more thoroughly discuss the evolution of usage of IOP-lowering drugs in the treatment of uveal melanoma patients; and also changes in the follow-up of these patients.

REPLY and Change: In the Methods-section paragraph 2.4 last paragraph we discussed the development/availability of the different classes of IOP-lowering medication in The Netherlands. As the duration of usage of IOP-lowering medication was not known, we calculated follow-up time by subtracting the date of diagnosis of UM with the date of metastasis/mortality. For controls, if applicable, we excluded patients with a baseline date before the date of launch of the respective IOP-lowering medication. Therefore, we performed for each class of IOP-lowering medication separate analyses (see Supplementary Table S1-9). As a consequence, the analyses with alpha2-agonists (i.e. the last launched class of medication: year 2000) had the smallest number of cases/controls totaling 589 patients. Thus (707-589=) 118 patients had a baseline date before 2000 and were excluded in the alpha2-agonists-analyses. Please see the last paragraph of the Methods-section paragraph 2.4.

The discussion section would also benefit from a better initial paragraph, as well as, a better organization of the discussion flow.

REPLY and Change: As requested we have re-organized the Discussion-section. We first discuss the main findings. Next, we discuss the prostaglandins and possible mechanisms, Subsequent, we discuss the mechanisms of lowering of IOP by the IOP-lowering medications groups (i.e. increase outflow or decrease production of aqueous humor). Then, we discuss the IOP itself and possible confounding-by-indication. Thereafter, we discuss possible (genetic) confounding factors. Finally, we mention the limitations of the study. Changes have been made through the Discussion-section.

In the supplementary material table the authors should clarify the presence of p-values of 1.000 in comparisons involving groups which have 0 patients displaying a specific feature (for example, positive family history for UM in table S1). Does it make sense to report a p-value in these cases?

REPLY and Change: We agree with the reviewer that a p=1.000 does not make sense. Therefore, we changed this to “Not applicable” (NA) in all analyses in which both analyzed groups had 0 patients. All analyses with such small groups were analyzed using Fisher’s Exact test.

The p=1.000 from the Fisher’s Exact tests follows from the following: For the Fisher’s Exact test multiple 2x2 tables are created, in which the row and column sums (based on the observed 2x2 table) are fixed. This results in a certain number of (expected) 2x2 tables. For each (expected) 2x2 table an odds ratio is calculated. If the odds ratio of one these (expected) 2x2 tables in similar the observed odds ratio, the probability of a deviation of at least the difference from an odds ratio of 1.0 is sure (i.e., true for all possible 2x2 tables). This is why the p=1.000.

Some authors/investigators choose not to report these p-values. We are not sure what the guidelines of this journal are regarding these p-values; however, if the suggestion is to remove them we are happy to do that.

Therefore, if considered relevant for publication in Cancers, the authors will have to address the aspects highlighted above before the final format of publication is achieved.

REPLY: We addressed all issues as pointed-out by the reviewer.

Reviewer 4 Report

From the point of view of the study design and statistical analysis plan (tests, correction for possible confounders, etc.) nothing to say. The problem is actually the low sample size for many groups in comparison: often it does not reach 5 units and therefore, rightly, the authors do not report results because they are considered methodologically unreliable. Where statistical tests confirm significance (P <0.05) an "effect" cannot be denied, but if the sample size is very low, below 10 units, a suspicion of significance of that group of patients arises. The authors honestly admit that the findings need confirmation from larger case studies (or longer follow-up so that more cases can be observed). So, from a general point of view, concluding that topical prostaglandin or pilocarpine may increase the risk of metastatic UM and mortality with this very small sample size is very risky for the authors, the journal and the overall patients management.
I think is mandatory a detailed analysis of the data by an expert statistician before thinking to publish this paper, mainly because, from a pure oncological point of view, thinking that drugs that increases the AH exit from the eye may enhance the metastatic potential of UM is similar to thinking that drugs that increase
glomerular filtration rate correlates an increased risk of renal metastasis from lung cancer.

Author Response

Reviewer 4:

Comments to the Author

From the point of view of the study design and statistical analysis plan (tests, correction for possible confounders, etc.) nothing to say. The problem is actually the low sample size for many groups in comparison: often it does not reach 5 units and therefore, rightly, the authors do not report results because they are considered methodologically unreliable. Where statistical tests confirm significance (P <0.05) an "effect" cannot be denied, but if the sample size is very low, below 10 units, a suspicion of significance of that group of patients arises. The authors honestly admit that the findings need confirmation from larger case studies (or longer follow-up so that more cases can be observed). So, from a general point of view, concluding that topical prostaglandin or pilocarpine may increase the risk of metastatic UM and mortality with this very small sample size is very risky for the authors, the journal and the overall patients management.

I think is mandatory a detailed analysis of the data by an expert statistician before thinking to publish this paper, mainly because, from a pure oncological point of view, thinking that drugs that increases the AH exit from the eye may enhance the metastatic potential of UM is similar to thinking that drugs that increase glomerular filtration rate correlates an increased risk of renal metastasis from lung cancer.

REPLY: As requested by the reviewer we approached a statistician who thoroughly reviewed the statistical analyses (see Acknowledgement-section). There were no flaws, but we agree that based on just one study with such small numbers of cases it is impossible to state that IOP-lowering medication increases the risk of metastasis and mortality. Nonetheless, the current findings suggest that IOP-lowering medication at least may affect the risk of metastasis and mortality in UM patients. Off course, more research is required to verify the revealed association and its pathophysiological mechanisms. Changes has been made to the Conclusions-section.

Round 2

Reviewer 1 Report

The authors have adequately replied to the issues raised and have added the information required. 

Reviewer 3 Report

In the revised manuscript "The effect of intraocular pressure-lowering medication on metastatic uveal melanomas" by Jan Pals et al. the authors aimed to address the important question on whether IOP-lowering drugs, employed to increases aqueous humor outflow, might lead to an enhancement of metastases development in uveal melanoma. The study comprises a large cohort of patients over a 35-year period. Through well conducted research, the authors found that the use of topical prostaglandin or pilocarpine is associated with an increased risk of metastases development and mortality in uveal melanoma. The paper is well written and organized and the theme is interesting and relevant, with their findings having important implications in the management of patients who are afflicted by this potentially deadly disease. The authors addressed in a very satisfactory manner my previous comments and suggestions. Therefore, I strongly recommend the publication of the manuscript in Cancers.

Reviewer 4 Report

Sorry, for me the conclusion are not supported by the results obtained using a control group of 6-12 persons. 

REPLY: We can not increase the number of cases. The control group includes >100 persons (and not 6-12). We mentioned this limitation in the Discussion-section. The conclusion is supported by the results; however, due to the low number of cases and upon request of the reviewer we changed the Conclusion (section 5) to make the statement of the revealed associated less stringent.